**Perspective**

# Liquid-liquid crystalline phase separation of spider silk proteins
**Michael Landreh** [1,2] ✉, **Hannah Osterholz** [1], **Gefei Chen** [1,3], **Stefan D. Knight**[1], **Anna Rising** [3,4] ✉ & **Axel Leppert** [1,2] ✉

Liquid-liquid phase separation (LLPS) of proteins can be considered an intermediate solubility regime between disperse solutions and solid fibers. While LLPS has been described for several pathogenic amyloids, recent evidence suggests that it is similarly relevant for functional amyloids. Here, we review the evidence that links spider silk proteins (spidroins) and LLPS and its role in the spinning process. Major ampullate spidroins undergo LLPS mediated by stickers and spacers in their repeat regions. During spinning, the spidroins droplets shift from liquid to crystalline states. Shear force, altered ion composition, and pH changes cause micelle-like spidroin assemblies to form an increasingly ordered liquid-crystalline phase. Interactions between polyalanine regions in the repeat regions ultimately yield the characteristic β-crystalline structure of mature dragline silk fibers. Based on these findings, we hypothesize that liquid-liquid crystalline phase separation (LLCPS) can describe the molecular and macroscopic features of the phase transitions of major ampullate spidroins during spinning and speculate whether other silk types may use a similar mechanism to convert from liquid dope to solid fiber.

Spider silk showcases a unique blend of strength, extendibility, and versatility tailored for various purposes including prey capture, shelter construction, and reproduction. With properties surpassing those of synthetic materials, spider silk holds promise for a wide range of applications, from biomedicine to advanced materials engineering. Understanding the structure and production mechanisms of spider silk is key for the successful production of silk-based biomaterials[1]. Spider silk is composed of proteins called spidroins, which vary in sequence depending on silk type. Spidroins contain a long region composed of repetitive low-complexity sequences that alternate between helical polyalanine blocks and disordered glycine, tyrosine, and serine-rich spacers[2,3]. The repeat domain is capped by small, α-helical domains at the N- and C-termini (NT and CT, respectively)[4]. In the spider, spidroins are expressed as soluble proteins by epithelial cells and secreted into the sac of the spinning gland, where they are stored as a fluid dope at concentrations exceeding 50% w/v[2,5,6]. When the dope is pulled from the sac through the spinning duct, it experiences a drop in pH from pH 7.4 to below pH 5.5, alongside increased shear force and reduced concentration of sodium and chloride ions, and increased concentration of potassium, sulfate, and phosphate ions[5,7,8]. Together, these factors induce a transition from soluble protein dope to insoluble protein fiber during which the polyalanine repeat regions convert to β-sheet blocks that determine the silk's properties, such as tensile strength[9,10]. All domains (NT, CT, and repetitive region)

likely act as specific sensors for pH and salt changes, with the monomeric NT domain undergoing dimerization and conformational stabilization, the repetitive region localized compaction and conformational rearrangements, and the constitutive CT domain dimer undergoing aggregation[11–15]. To facilitate the fast transition from dope to fiber, spidroins must be highly aggregation-prone. In fact, recombinant production of even short repeat domains in a variety of expression hosts has yielded only insoluble material[1], which raises the question how spiders control spidroin solubility in the gland.

In this perspective, we summarize current evidence for the existence of different solubility regimes during spinning. We suggest that the silk dope undergoes a transition from liquid–liquid to liquid–crystalline to crystalline state, brought about by competing interactions in the repetitive and non-repetitive sequences of the spidroins. Sequence analysis of different spidroins suggests that this transition may be a common feature in all silk types.

## Liquid–liquid phase separation of fiber-forming proteins

Unlike folded proteins, which can self-chaperone by constraining aggregation hotspots within their structures[16], disordered proteins undergo rapid conformational changes that can expose aggregation sites. There is rapidly accumulating evidence that disordered proteins can also populate an

[1]Department of Cell and Molecular Biology, Uppsala University, Uppsala, Sweden. [2]Department of Microbiology, Tumor and Cell Biology, Karolinska Institutet, Solna, Sweden. [3]Department of Medicine Huddinge, Karolinska Institutet, Huddinge, Sweden. [4]Department of Animal Biosciences, Swedish University of Agricultural Sciences, Uppsala, Sweden. ✉e-mail: Michael.Landreh@icm.uu.se; Anna.Rising@slu.se; Axel.Leppert@icm.uu.se

intermediate solubility regime by engaging in liquid–liquid phase separation (LLPS)[17]. The hallmark of LLPS is the formation of droplets ranging from sub-micrometer to several micrometers in size, which appear liquid yet are clearly separated from the surrounding solution, resembling oil droplets in water[18]. The prevalent model for LLPS includes contacts between short sequence motifs (stickers) that are separated by disordered regions (spacers)[19]. In the resulting assembly, sticker contacts are frequently broken and re-form due to their relatively low affinity, while spacer flexibility allows for frequent conformational rearrangements. Often, stickers and spacers form low-complexity sequences, with alternating patches of amino acids that can interact with each other, and amino acids that enhance flexibility. Common stickers are aromatic, polar, and charged amino acids[20–22]. Aromatic amino acids such as phenylalanine, tyrosine, and tryptophan participate in π-stacking interactions but also contribute to the formation of hydrophobic clusters within the protein. Polar residues, particularly serine and threonine, engage in hydrogen bonding interactions with water molecules or other polar residues, influencing the solubility and phase behavior of the protein. Charged residues like arginine, lysine, glutamate, and aspartate play a crucial role in electrostatic interactions, which can drive LLPS by exerting attractive or repulsive forces between molecules. Positively charged residues form salt bridges with negatively charged residues but also engage in cation-π interactions with aromatic residues, considered one of the major sticker interactions in LLPS (Fig. 1)[21]. Glycine, which lacks a side chain, is enriched in spacer sequences and promotes LLPS by enabling conformational flexibility and close packing of protein molecules. While proline is less commonly associated with directly promoting LLPS, its presence in low-complexity sequences enhances conformational dynamics and assembly properties[23].

Evidence is rapidly accumulating that LLPS and fiber formation are related phenomena[24]. Many disordered proteins and peptides that form amyloid fibers in vivo can also undergo LLPS, for example the Parkinson's-associated α-synuclein protein, amyloid-β peptide, which is implicated in Alzheimer's disease, and TDP-43 and FUS, fibrillar aggregates of which have been found in patients with frontotemporal dementia[24–26]. While LLPS and fiber formation may be inherent properties of proteins that can be triggered by specific solution conditions[25–28], the repetitive nature of many low-complexity sequences favors the formation of steric zippers that are a hallmark of protein fibers (Fig. 1)[29,30]. Here, random contacts between disordered stickers can give way to in-register alignment of the sequences and eventually fibrillation. In fact, droplets composed of proteins with low-complexity sequences can be converted to fibers near-instantly by applying shear force[31]. While this process is associated with the formation of pathogenic protein aggregates, it is likely equally relevant for functional amyloids, i.e., proteins whose self-assembly into β-sheet-rich structures such as fibrils and fibers serves a biological purpose[32,33].

## Liquid–liquid and liquid–crystalline interactions in spidroins

Functional amyloids must be able to assemble into fibers on cue[32,34]. In the case of spidroins, this challenge is exacerbated by the need to remain non-fibrillar at the extreme concentrations inside the silk gland's sac. Early studies on the content of silk glands revealed that the spinning dope shows birefringence when squeezed between two glass slides and viewed under polarized light, indicating a liquid crystalline solution[35,36]. In a liquid crystalline state, proteins are assembled into partially ordered structures, but their interactions are weak enough to remain liquid-like. This idea received further support from reports that the dope could be converted to fibers through shear, which further aligns fibrillar components and reduces water content. On the other hand, the spidroin dope is not a homogeneous liquid crystal solution, as secreted spidroins in the silk gland appear as granular structures that are several mm in diameter, potentially forming micelle-like structures[3,37]. NMR and electron microscopy of native dope isolated from the silk gland suggested that the repeat regions of multiple spidroin monomers can entangle into highly disordered, dense clusters with the terminal domains potentially oriented towards the surface[38,39]. These clusters measure 20–25 nm in diameter and interconnect into a network of mm-sized assemblies that resemble the granules observed in silk glands[39,40]. In line with this model, native mass spectrometry showed that recombinant spidroins adopt compact conformations with a similar density as globular proteins[41]. Recombinant and native spidroins can form spherical droplets in vitro when exposed to high concentrations of kosmotropic ions that cause compaction and self-association of the repeat domains[42–44]. The droplets show hallmarks of LLPS like the presence of a threshold concentration for assembly, droplet fusion, and fluorescence recovery after photobleaching, as

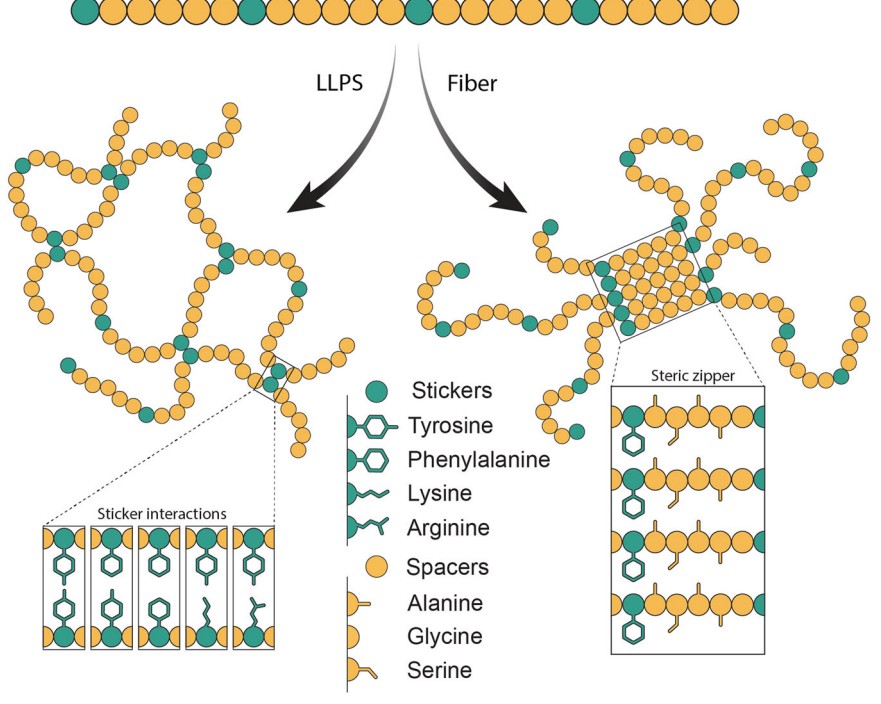

**Fig. 1 | Low-complexity domains contribute to LLPS and fiber formation.** A typical phase-separating low-complexity sequence can be divided into stickers (green) and spacers (yellow), which provide low-affinity contacts and flexibility, respectively, to mediate LLPS. Similarly, low-complexity sequences can assemble into energetically favorable steric zippers that exclude water from their interface and form stable fibrillar aggregates. Here, low complexity facilitates easy alignment of complementary residues. For clarity, the ratio and distances of stickers and spacers have been simplified.

assessed by turbidity measurements, light and fluorescence microscopy, and rheology[42,44,45].

What sequence features mediate phase separation of spidroins? Major ampullate spidroins 1 and 2, which are the main component of the dragline silk[46], are among the most-studied spider silk proteins, and homologs from different species have been utilized in most studies demonstrating LLPS[42,45,47,48]. The major ampullate spidroin contains folded N-terminal and C-terminal domains connected by a disordered repeat region (Fig. 2). The C-terminal domain in synergy with the repeat region has been identified as the main driver for LLPS while the N-terminal domain has no effect on LLPS but accounts for the high solubility of spidroins[42,49,50]. The repeat region of MaSp1 spidroins is composed of two alternating types of sequences: alanine homorepeats that can be 6–14 residues in length[51], and glycine-rich segments of 10–20 residues that contain glutamine, tyrosine, serine, and arginine residues. The roles of these two types of sequences in the spun fiber have been described in detail elsewhere[2,9,52]. Briefly, the polyalanine regions can be helical or disordered in the soluble state, depending on repeat length[53], but are packed β-strand crystals in the assembled fiber, where they contribute to fiber strength[54,55]. The glycine-rich regions remain disordered in both soluble and fibrillar spidroins and are associated with super-contraction and fiber elasticity[9]. Importantly, the glycine-rich regions share the amino acid composition of low complexity sequences in many phase-separating proteins: glycine and serine are associated with flexible spacers, while arginine and tyrosine are among the most common types of stickers[21,23]. Indeed, mutating the arginines or tyrosines in the repeat regions drastically changes the ability of mini-spidroins to undergo LLPS (Fig. 2)[48].

The CT domain has also been found to contribute to LLPS of mini-spidroins, despite being a folded domain without any LLPS-associated sequence features[42]. However, the fact that CT is a constitutive dimer means that it effectively doubles the number of the attached repeats, which may bring the protein across the threshold for LLPS. In addition, minor ampullate spidroins contain a non-repetitive, disordered sequence right before the C-terminal domain which is also rich in sticker and spacer residues[48]. LLPS of this region causes destabilization of the native CT domain dimer, which could serve as a primer for CT aggregation[48]. These data suggest that the glycine-rich domains mediate major ampullate spidroin LLPS via the stickers and spacers model. However, homorepeats of single amino acids can also promote phase separation. Polyalanine was recently shown to undergo LLPS due to its balance of secondary structure propensity and hydrophobicity[56,57]. Importantly, its native α-helical structure is destabilized at increasing repeat length, but also when increasing numbers of polyalanine repeats associate with each other[56]. Under these

conditions, polyalanine favors packing into β-strand crystals via steric zippers (Fig. 2).

Together, the evidence paints a Janus picture of the state of spidroins: On one hand, they contain glycine-rich regions that favor phase separation via stickers and spacers. On the other hand, they contain polyalanine regions that favor crystalline aggregation. These competing properties are obviously related to the requirement to be produced in soluble form yet rapidly assembled into solid fibers. But how do spidroins convert from a liquid to a solid phase? We hypothesize that this transition can be viewed through the lens of the liquid–liquid crystalline phase separation (LLCPS) model of biomolecular solutions[58]. In this model, the macroscopic assembly contains liquid crystal-forming components, such as DNA, and non-liquid crystal-forming components, for example, polycations. These components are usually separate molecules that can either co-associate into a dense phase or segregate with one component as dilute and one as dense phase. The resulting co-assemblies balance between a more liquid and a more solid state. The state is determined by factors like shear, pH, protein concentration, ionic strength, ion composition, and temperature, depending on whether they promote interactions between liquid crystal components, which drive crystallization, or between liquid crystal- and non-liquid crystal components, which favor LLPS. This model has explicitly been applied to amyloidogenic proteins[58]. Importantly, the LLCPS model allows for regulation of the solubility state of its components, the key feature of the silk spinning process. We would like to point out, however, that this hypothesis has not been addressed experimentally, for example by defining the mesogen, the liquid crystal forming unit, of spider silk. Micelle-like structures composed of amyloidogenic protein repeats have been suggested as the mesogen of silkworm fibers, which may be transferrable to spidroin assemblies formed via LLPS[59].

## Evidence for LLCPS spinning of spider silk
Multiple studies suggest that spherical structures represent an intermediate step, but also a dead end in the silk fiber formation pathway, depending on the assembly conditions[43,45]. Following the spinning process in situ is challenging, but engineered mini-spidroins and microfluidic systems that recapitulate key steps -acidification and shear- have offered important clues to its mechanism[1], Preparation of native-like spinning dope composed of CT-containing spidroins and high phosphate concentrations resulted in droplet formation[60] and fibers with improved toughness[61], underscoring that phase separation is a natural component of silk formation. Furthermore, non-native silk dope composed of repeat domains in denaturing solvents underwent LLPS in response to increased elongational flow in a

---

**Fig. 2 | LLPS and fiber formation are mediated by separate regions of the repeat region in major ampullate spidroins. a** Architecture of a major ampullate spidroin, which contains folded N-terminal and C-terminal domains (purple and orange, respectively), connected by a disordered repeat domain with alternating polyalanine blocks (red) and segments rich in glycine, tyrosine, serine, and glutamine (blue). **b** The NT domain dimerizes in an antiparallel orientation, regulated by pH changes and complementary surface electrostatics. **c** Polyalanine sequences adopt a helical or random coil conformation in small bundles but convert to β-strands as more polyalanine segments are added. Shown are AlphaFold predictions for three to six copies of a 15-alanine stretch as found in MaSp1. **d** Y and R sticker residues in the repeat regions mediate LLPS. Changing the high-affinity sticker tyrosine to the lower-affinity sticker phenylalanine increases LLPS while replacing the high-affinity sticker arginine with leucine, which has very low LLPS propensity, reduces LLPS (adapted from[48]).

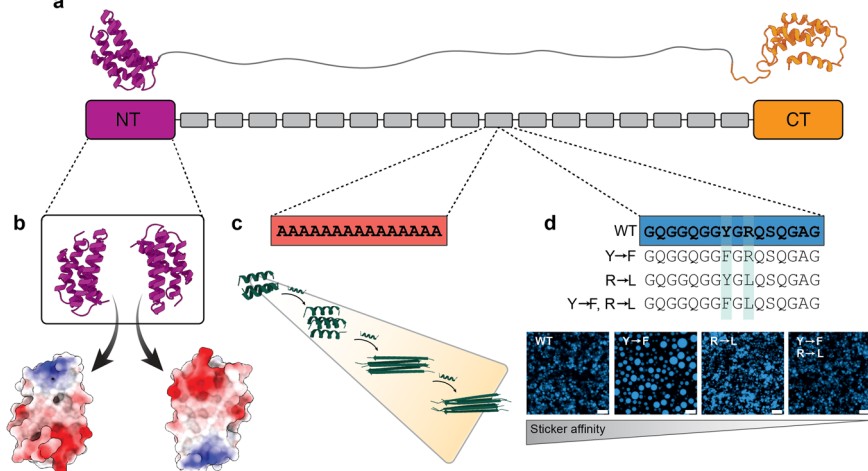

microfluidic spinning apparatus[62]. Similarly, microscopy of concentrated solutions of the mini-spidroin NT2RepCT, which contains two repeat domains flanked by the NT and CT domains[63], showed concentration-dependent droplet formation during extrusion in a biomimetic spinning set-up[64]. This observation could be corroborated by microscopic and spectroscopic analysis of silk dope isolated from different parts of the silk gland, showing micrometer-sized granules that are aligned and fused during passage through the duct, while their structural order increases[40]. The mechanism was subsequently reproduced in a microfluidic device in which the dope was exposed to a concentrated phosphate buffer as LLPS trigger prior to initiating fiber formation through a pH drop and shear[65]. In this system, which is possibly the most biomimetic to date, droplet formation, shear, acidification, and importantly number of polyalanine-containing repeat regions were found to synergistically increase the β-sheet content. Similarly, NMR studies of silk spun with and without prior phase separation revealed differences in the local environment of the repeat domain tyrosines, linking silk structure to sticker interactions in the droplet state[47].

What is the role of phase separation in silk storage and assembly? In trying to answer this question, we must consider the structure of the silk dope, how it relates to the interactions of the individual spidroin domains, and how those interactions are affected by shear, ions, and pH (Fig. 3a–c). In the sac of the gland, silk is stored as micrometer-sized liquid droplets, likely aided by balancing the concentration of different ions[8,37,42,63]. The droplets appear homogeneous, but electron microscopy suggests that the liquid state involves a dense network of micelle-like structures and water-filled pockets[39]. Assembly is driven by weak interactions between tyrosine and arginine stickers embedded in flexible glycine-rich regions[42,48]. We speculate that in the dynamic environment of a liquid droplet, self-association tendencies of the polyalanine repeats may be mitigated. Upon entry into the duct, the ion composition gradually changes from chaotropic (Na, Cl) to kosmotropic (SO$_4$, PO$_4$), which favors the formation of intermolecular interactions[10,44]. At the same time, the dope is dehydrated and sheared. Cryo-electron microscopy shows that the micelle-like structures become more elongated under these conditions[39]. Simultaneous acidification crosslinks the micelle-like structures via their NT domains and initiates the structural conversion of the CT domain (Fig. 3b)[5,11]. At the atomistic level, the shift from LLPS to LC is accompanied by structural changes in the repeat domain. The low-affinity "stickers and spacers" contacts give way to assembly of polyalanine segments into steric zippers which eventually give way to the solid state of the spun fiber (Fig. 3c).

We can describe the above steps in terms of liquid crystal phase transitions (Fig. 3d). During storage, LLPS according to the "stickers and spacers" model keeps the silk in a liquid state. The formation of micelle-like mesogens in the liquid droplets gives rise to an isotropic liquid crystal phase. Shear and pH elongate and align the mesogens, resulting in a nematic liquid crystal phase. In this phase, tactoids nucleate via micro crystallization of steric zipper sequences that may become the crystalline blocks in the dried silk. Hence, the spidroin dope might be described as LLCPS, essentially shifting downward through the phase diagram from liquid to liquid crystalline to crystalline (Fig. 3e).

## Does LLCPS apply to other silk types?

As outlined above, the presence of two types of sequences in the repeat domain could potentially mediate a shift from liquid to crystalline states during spinning: LLPS-promoting stickers and spacers drive mesogen formation through LLPS, and polyalanine zippers drive the conversion of mesogens to solid fibers. However, this model is based on the major ampullate spidroins, which are well-studied, while comparatively little is known about storage and spinning of other silk types. Major ampullate, minor ampullate, tubuliform, and flagelliform spidroins contain NT domains that undergo pH-dependent dimerization[66]. The CT domain and its pH-dependent self-assembly are similarly conserved between spider species[11] and among major ampullate, minor ampullate, aciniform, and tubuliform spidroins[67]. These correlations indicate that the pH-dependent aspects of the assembly process are similar for all these silk types, which

raises the question whether LLCPS also is a common feature of the silks. Indeed, the repeat domains from tubuliform spidroin 1 and minor ampullate spidroin 1 undergo phase separation in vitro[62].

As there is little experimental data on the biophysical and biochemical properties, such as LLPS, of other silk types than major ampullate spidroins, we predicted the LLPS propensity of 200 amino acids from the repeat regions from five different silk types from *A. ventricosus* (major ampullate spidroin 1 and 2, minor ampullate spidroin, flagelliform spidroin 1, aciniform spidroin, tubuliform spidroin 1, and pyriform spidroins 1 and 2) using the FuzDrop server (Fig. 4)[68]. We additionally predicted the occurrence of aggregation hotspots, that is, sequences with a high propensity to form steric zippers, a prerequisite for the formation of β-sheet-rich fibers[16]. Strikingly, all repeat domains contain extensive LLPS-promoting regions (Fig. 4). As expected for major ampullate spidroin, we found a constant LLPS propensity of nearly 100%, with small dips at the sites of the self-assembling polyalanine segments. We made similar observations for flagelliform spidroin 1, where the aggregation hotspots are composed of valine and isoleucine-rich stretches. In pyriform spidroin 2, the alanine segments are disrupted by di-glutamines, which lowers their aggregation propensity slightly compared to polyalanine alone, however, the repeat region structure of this spidroin is highly complex and likely combines several self-assembly mechanisms[69]. In all these cases, LLCPS appears a possible mechanism for liquid-to-fiber conversion. For the other repeat domains, a different picture emerges. Minor ampullate spidroin showed perfect LLPS scores, in line with the absence of homorepeats and an even distribution of glycine, serine, alanine, and tyrosine. It includes folded "spacer" domains that are prone to form fibrillar aggregates[70]. In line with the suggestion that this spidroin is closer to LLPS than a liquid crystalline state, the isolated repeat domain does form amyloid-like fibrils in vitro but no fibers unless connected to a CT domain[71,72]. The repeat domain of aciniform silk contains a folded domain, that has been shown to form fibrillar aggregates[72] and is flanked by an LLPS-promoting region. Recent studies have suggested that during spinning, the folded domains align and undergo partial refolding to β-strands, but do not appear to aggregate completely[73]. Similar suggestions were made for the repeat domain organization of pyriform spidroin 1 and tubuliform spidroin 1, both of which also contain helical domains as part of a "bead on a string" organization[74,75]. The high LLPS propensity that is predicted for the regions between the folded domains suggests a similar assembly mechanism as for major ampullate spidroins, in which contacts between the folded domains are regulated through LLPS of the disordered spacers. Importantly, we do not conclude that LLCPS is a common feature of spidroins but suggest that some of these proteins may exhibit a similar LLPS/liquid crystalline "Janus face" as the major ampullate spidroins.

## Outlook

In conclusion, the LLCPS model provides a potential framework for understanding the pre-spinning state of spider silk proteins. Major ampullate spidroins undergo a transition from liquid–liquid to liquid–crystalline states through interactions within their polyalanine regions, leading to the formation of β-crystalline structures in mature silk fibers. This dynamic process is tightly regulated by environmental factors such as pH, shear force, and ion concentration, ensuring efficient fiber formation. The LLCPS model may also apply to other types of spider silk, suggesting a common mechanism underlying silk assembly across different species.

Further studies are needed to confirm that both liquid–crystalline and liquid–liquid states coexist under spinning conditions. Our model suggests the formation of an LC mesophase via LLPS, which gives rise to crystalline domains. Since different sequence features mediate LLPS and crystal formation, the first open question is how each feature affects the other assembly type. Specifically, how do mutations of the stickers and spacers affect crystallization of polyalanine, and how does the length of the polyalanine repeat affect the phase separation behavior? Linder and co-workers recently showed that LLPS promotes the formation of dynamically arrested states in spidroins[76], which could support our hypothesis. This issue can be addressed

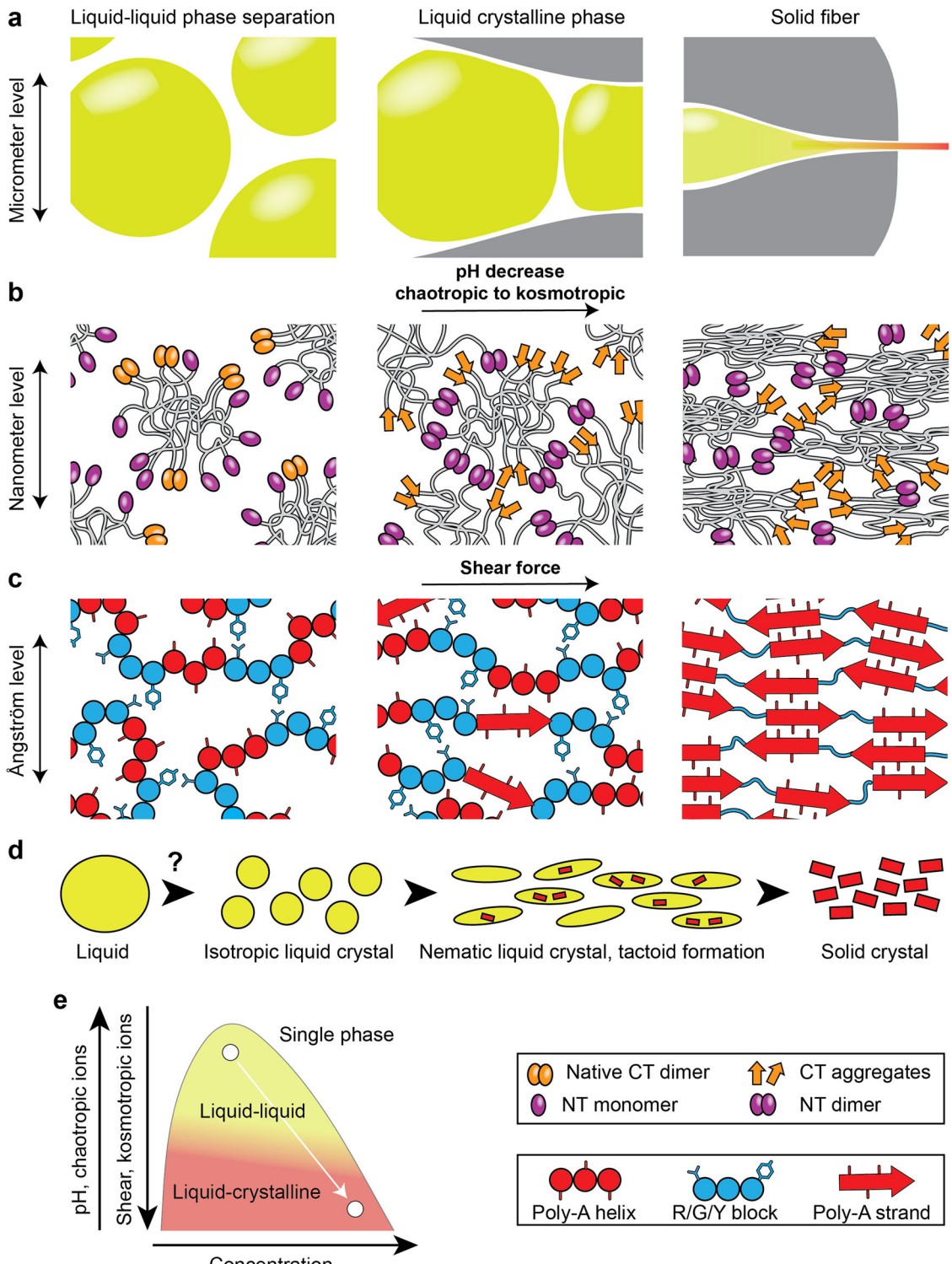

**Fig. 3 | A potential framework for LLCPS as an intermediate step of spider silk assembly, based on in vitro models of spidroin assembly. a** The silk dope forms liquid droplets in the gland (left). Spinning is initiated when droplets move into the spinning duct (middle). At the distal end of the duct, the dope appears as homogeneous fiber (right). **b** Inside the liquid spidroin droplets, micelle-like structures with terminal domains at the surface and repeat domains entangled in the core form the mesogen of the LC phase (left). Reduced pH causes dimerization of NT domains, linking individual assemblies, and unfolding and self-assembly of the CT domains (middle). The increased ordering of the mesogen shifts the character of the dope from LLPS to LC. The cross-linked micelle-like structures are compressed and dehydrated as they travel further through the duct (right). **c** In the core of the assemblies, the repeat regions engage in LLPS mediated by the glycine, serine,

tyrosine, and arginine-rich blocks, keeping the polyalanine sequences apart and likely helical (left). Shear force aligns and locally concentrates the polyalanine sequences to start β-strand formation and initiate crystallization (middle). These crystalline microdomains form the tactoids that promote the shift from LC to crystalline phase. In the spun silk, the polyalanine regions are assembled into crystalline regions connected by the glycine, serine, tyrosine, and arginine-linkers. **d** Liquid, liquid–crystalline, and solid crystalline phases are observed during silk spinning, following the progression from left to right in (**a**–**c**). **e** Sketch of the LLCPS phase diagram, indicating the shift from liquid–liquid to liquid crystalline phase separation due to decreased pH, altered ion composition, increased shear, and increased concentration.

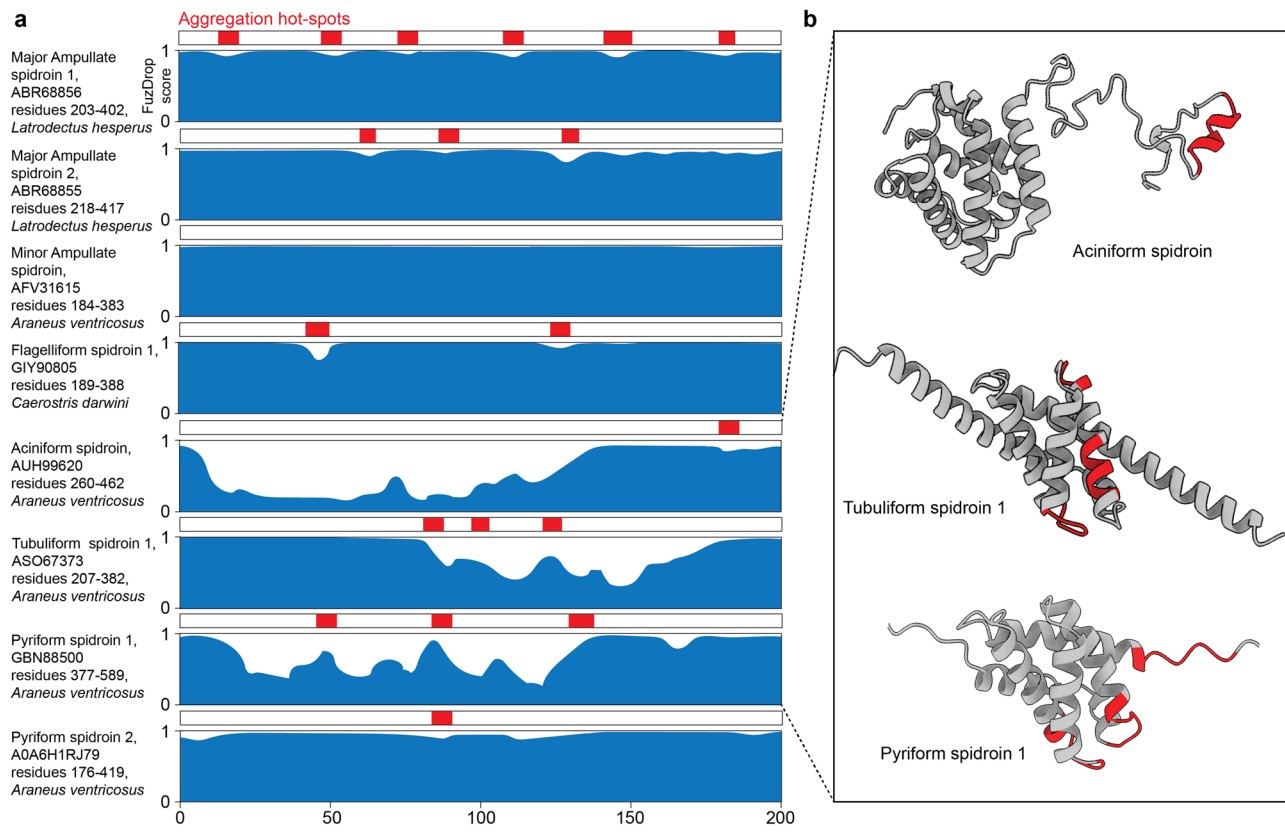

**Fig. 4 | LLPS- and aggregation-promoting regions in the repeat domains of different spidroins. a** The first 200 residues of the repeat regions of, top to bottom, major ampullate spidroins 1 and 2, minor ampullate spidroin, flagelliform spidroin 1 A, aciniform spidroin, tubuliform spidroin 1, and pyriform spidroins 1 and 2 were analyzed using the FuzDrop server[68]. Regions with a FuzDrop score above 0.9 are considered LLPS-promoting. Aggregation-prone regions are indicated in red. For the major ampullate spidroins, as well as flagelliform spidroin 1, regions with high β-strand potential coincide with a dip in the LLPS propensity, indicating the presence of separately encoded liquid–liquid and liquid–crystalline interactions. **b** AlphaFold structure predictions for the repeat sequences from aciniform spidroin as well as tubuliform and pyriform spidroin 1 suggest that the non-LLPS regions identified by FuzDrop represent folded domains. Aggregation-prone sequences (red) are buried in folded structures. The aciniform structure prediction is in excellent agreement with the NMR structure the aciniform repeat from another species[73].

by engineering recombinant mini-spidroins and assessing their phase separation properties and β-sheet content with biophysical methods.

The second question relates to the conversion from nematic crystal phase to solid fiber. Would the formation of crystalline microdomains propagate to other polyalanines in the mesogen? Additional shear could promote the formation of larger crystals by aligning the microdomains[77], but recent studies suggest that not all polyalanines are converted to β-strands[38]. Answering this question would require highly time-resolved structural studies of spidroins under assembly conditions. The fact that liquid spidroin droplets can be converted to gel by laser pulses provides a possible window for spectroscopic investigations of the crystallization process[78].

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

## Acknowledgements

M.L. is supported by a KI faculty-funded Career Position, a Cancerfonden Project grant (22-2023 Pj), and a Consolidator Grant from the Swedish Society for Medical Research (SSMF). A.L. is supported by the Olle Engkvist Foundation (to ML). A.R. is supported by the European Research Council (ERC) under the European Union's Horizon 2020 research and innovation program (grant agreement No 815357), Olle Engkvist Foundation (207-0375), the Center for Innovative Medicine (CIMED) at Karolinska Institutet and Stockholm City Council, the Swedish Research Council (2019-01257) and Formas (2019-00427).

## Author contributions

A.L., A.R., and M.L. designed the study. H.O., G.C., and S.D.K. prepared images, analyzed data, and drafted sections. A.L. and M.L. compiled the manuscript with input from all authors.

## Competing interests

The authors declare no competing interests.
