## [Peer review file · Communications Chemistry]

Liquid-liquid crystalline phase separation of spider silk proteins

Corresponding Author: Professor Michael Landreh

Version 0:

Reviewer comments:

Reviewer #1

(Remarks to the Author)

This manuscript aims to bring a new framework for understanding spidroin organization and self-assembly during the spinning process. In particular, it seeks to bridge the two theories that have dominated our contemporary understanding of spider silk spinning: the liquid crystal theory and the micelle theory. Both of these theories are supported by experimental observations, yet they have, until now, seemed incongruous. Thus, a framework that connects these two theories would provide an impactful foundation for guiding future research in this field.

This manuscript is expertly written. Generally, the references provide a thorough and up-to-date catalog of publications on the topic. The figures are well-made and clearly illustrate main concepts. If published, I am certain that my group would frequently cite this paper in our work.

However, there are several points that need additional clarification or discussion before publication:

1) The role of alanines, in particular the polyalanine segments in spidroins, is not clearly presented. Lines 73-74 do not include alanine as a "sticker", yet on line 146, alanine is referred to as a "sticker". If it is a sticker, by what mechanism is it sticky? Figure 1 shows alanine to be in the "flexible" spacer region, not the sticker region; however, I would not consider polyalanine to be "flexible" even if it is in a helical conformation. I assume that the polyalanines constitute the steric zippers that the authors refer to, but this is not clearly written in the manuscript. Based on my reading of the section "Does LLCPS apply to other silk types?", it seems like the authors mean to suggest that the polyalanine is neither a sticker or a spacer but should be considered separately as not part of the LLPS domain.

2) In line with my comment #1, the mapping between stickers/spacers (LLPS theory) and hard(crystallizing)/soft repeating domains (LC theory) is still not clear.

3) The authors refer to the polyalanines as ~15 residues long, with alpha-helical structure in the pre-spinning state. However, many polyalanine segments in spidroins are shorter than 15 residues. For example, ADF-4 has polyalanines that are approximately 6-12 residues long. At these lengths, the segments have less helical propensity and more random coil propensity (<https://pubs-acscs-org.libproxy.rpi.edu/doi/10.1021/bi201155g> and others). This discussion should be added to the manuscript.

4) Lines 198-199 describe phenylalanine as a lower affinity sticker compared to tyrosine, which increases LLPS. It's not clearly explained why phenylalanine is less sticky than tyrosine, and why lower stickiness increases LLPS. Similarly, it's not clear to me why Leu is a higher affinity sticker.

5) On line 232, the authors speculate that "LLPS helps to keep the poly-alanine regions from spontaneous alignment". However, it's not clear to me why polyalanine regions would spontaneously align. This may be a wording choice. Polyalanine segments may spontaneously form beta-strands that are locally aligned on the angstrom scale, but without shear or elongational flow, they should not universally align along any particular axis.

6) On line 239, the authors state "Hence, the pre-spinning state might be described as LLCPS...". This is a bit unclear to me, as no crystallization occurs (according to the authors on lines 234-235) until the dope enters the spinning duct.

7) Lines 181 - 183 state that "Further studies are therefore needed to confirm that both liquid-crystalline and liquid-liquid states co-exist under spinning conditions". These studies seem like they should be straightforward, so a discussion of why such studies have not been performed (challenges and barriers?) would enhance the manuscript. Additionally, more

discussion on what is known about the liquid crystalline states, and how that may conflict or coincide with LLPS states would be helpful. For example, how might a nematic liquid crystal phase "fit" into a spidroin-rich LLPS domain? Presumably, LC domains would be much smaller and therefore reside within the LLPS domains?

8) on line 142, the authors state that the polyalanines "account for fiber strength". This seems to be an oversimplification, as the beta-sheet nanocrystals also contribute to stiffness and toughness. (<https://www.nature.com/articles/nmat2704> and <https://doi.org/10.1002/adfm.201600813> may be good references)

Reviewer #3

(Remarks to the Author)

Previous studies have reported that major spidroins can be stored in the silk gland (SG) through liquid-liquid phase separation (LLPS) before undergoing spinning. The author provides an in-depth overview of the transition from the liquid-liquid phase to the solid state experienced by ampullate spidroins within the spider silk gland. From a sequence perspective, the highly repetitive poly-alanine and glycine-rich regions play crucial roles in this process, with poly-alanine aiding in the formation of crystallized structures and glycine-rich regions mediating LLPS. The review also discusses various environmental factors that promote phase transition and trigger the C/N terminals to form amyloid-like fibrils. Lastly, the author compares the highly conserved repetitive motifs in different spidroins and suggests that the LLPS model may be a universal mechanism for silk assembly in spiders and potentially in other fibers produced by insects. This review is well-organized and good at illustrating. I recommend the publication of this review after minor revisions:

1. In Figure 1, I suggest including both interchain interactions (such as anti-parallel β -sheets) and intrachain interactions. This will provide a more comprehensive understanding of the structural dynamics during silk assembly.
2. It is beneficial to offer a more detailed illustration of the dimerization of the N-terminus through electrostatic interactions.
3. What are the specific ion species mediating spider silk glands' phase transition? This could help clarify the molecular mechanisms involved in the transition from LLPS to a solid state.
4. On page 6, the author summarized factors influencing the state of spidroins, stating that "The state is determined by factors like concentration, ionic strength, and temperature, depending on..." I recommend adding pH and shearing force to this list, as these are also critical parameters that could influence phase transitions during silk formation.

Reviewer #4

(Remarks to the Author)

The review manuscript by Michael Landreh et al. focusses on the transition process of spider silk proteins from soluble dope state to solid fiber formation. Specifically, they evaluate available literature on the role of liquid-liquid phase separation in the fiber spinning process, and propose a potential model describing a shift from liquid-liquid to liquid-crystalline state of the silk protein dope (Major Ampullate silk). As a conclusion, the authors hypothesize that the concept of liquid-liquid crystalline phase separation (LLCPS) – as described in literature for other biological systems – can be adopted to major ampullate spidroins, as well as to specific other silk types.

The review integrates general concepts of phase separation phenomena and fiber assembly discussed in the (spider) silk community, and aims to develop a detailed but rather intergrating view of the interacting aspects in major spidroin fiber formation, based on available literature and experimental data. As an interesting add-on, a comparison of five different silk types from an orb web spider based on predicted LLPS propensity is included to check if LLPS/LLCPS behaviour could be a general feature of silk spidroins.

The structured, well-written paper is of high interest to other researchers in the silk community, but also for the wider field of (fibrous) biopolymers.

However, there are several aspects in the manuscript that need to be addressed or clarified:

1) The main concept of the authors is the combined view of LLPS and liquid crystal state of the silk spidroin, leading to the proposed LLCPS behaviour. This is mainly introduced in the paragraphs 1.163ff & 225ff, and the way the authors formulate their arguments could lead to a conceptual misunderstanding: A) crystal and crystallinity aspect: the crystalline β -sheet formation of Poly-A stretches is not directly related to a liquid crystal state of a (bio)polymeric material (see also abstract), since the latter is defined as a bulk material state with specific properties, due to ordered molecules, or mesogens, that display liquid-like behaviour. Upon reading the manuscript, the reviewer struggled with the proposed view that β -sheet formation of the Poly-A stretches as crystallization leads to the liquid crystal state of the dope. B) Conceptual and sequential aspect: As cited, liquid crystal state has been proposed for silk worm silk by Walker et al. 2015; they argue it is based on micelle-like lyotropic mesophases, responsible for the orientation and properties of the dope. Thus, the micellar-like structures themselves seem to lead to liquid crystal state/behaviour, and "Solidification occurs concurrently with the structural transition to the final extended- β -sheet structure". Thus, shear stress, pH-induced conformational changes, ion conc. & composition changes, and increased intermolecular interactions in concert would lead to LLCPS and the formation of solid fibers as final result.

However, the reviewer does not question the proposed general concept of the LLPS and LLCPS behaviour of the silk spinning dope.

The authors are asked to clarify these aspects/rewrite text passages for better understanding.

2) Conceptual confusion or wording issue using the term aggregation in the manuscript: the reviewer disagrees with the view of Poly-A aggregation to β -sheet secondary structures. Instead, the term assembly should be used throughout, since protein aggregation is def. as spontaneous, uncontrolled structure formation. This applies even more so to aggregation of the CT-domain I.236 and Fig. 3. In the cited references e.g. Hagn et al. 2010, the pH-sensitive CT acts as a conformational switch due to exposure of hydrophobic domains upon acidification, allowing the two rep. core regions to get closer together and alignment of β -sheet forming repetitive sequence elements. There is no evidence of CT aggregation, but the dimer remains stable. It is highly unlikely that the concerted fiber spinning assembly incorporates uncontrolled aggregation mechanisms.

3) The role of ion conc. and composition is generally reflected in the manuscript, but it seems that the authors focus more on pH shift and shear stress as spinning factors, and reduce the role of ions in their text mostly to ion concentration changes. The role of ion exchange, i.e. composition of chaotropic and kosmotropic ions, has been established in the literature, and should be included in more detail in the manuscript to add clarity.

4) Another conceptual confusion or wording issue using the term micelle in the manuscript: micelles usually describe solvent-driven spherical structures with double-lipid layer. The silk community should use micelle-like, or micellar, structures of the spidroins instead for clarity.

5) In the paragraph on other silk types, it would be helpful if the authors may include both a more detailed discussion of the abundance of so-called "aggregation hot-spots" (presumably β -sheet forming motifs) in relation to the discussed LLCPS behaviour and fiber formation; in addition, it would be interesting to further discuss the reported results in relation to the various biological functions of the diff. silk proteins. It would be particularly interesting to discuss the differences in the two pyriform spidroins, as well as the consequences of the LLPS or LLCPS behaviour for this attachment cement.

Version 1:

Reviewer comments:

Reviewer #1

(Remarks to the Author)

This manuscript is well-written and presents an insightful perspective on phase transitions and macromolecular interactions in the silk spinning process. The authors have sufficiently addressed my critiques in this revision.

Reviewer #3

(Remarks to the Author)

The authors have addressed my previous concerns. I recommend this paper being published.

Reviewer #4

(Remarks to the Author)

The interesting review paper by Landreh et al. on "Liquid crystalline phase separation of spider silk proteins" in the revised version has been clearly improved, questionable scientific aspects and wording have been resolved by expanded and re-written passages & substitutes, and the general readability of the manuscript is improved. Most of my comments (and of the other reviewers) have been addressed in a conscientious manner, and I suggest that the revised manuscript may be accepted for publication as is. I will happily endorse spreading of the paper after its publication.

Response to Reviewers

Reviewer #1 (Remarks to the Author):

This manuscript is expertly written. Generally, the references provide a thorough and up-to-date catalog of publications on the topic. The figures are well-made and clearly illustrate main concepts. If published, I am certain that my group would frequently cite this paper in our work. We are grateful for the reviewer's appreciation!

However, there are several points that need additional clarification or discussion before publication:

1) The role of alanines, in particular the polyalanine segments in spidroins, is not clearly presented. Lines 73-74 do not include alanine as a "sticker", yet on line 146, alanine is referred to as a "sticker". If it is a sticker, by what mechanism is it sticky? Figure 1 shows alanine to be in the "flexible" spacer region, not the sticker region; however, I would not consider polyalanine to be "flexible" even if it is in a helical conformation. I assume that the polyalanines constitute the steric zippers that the authors refer to, but this is not clearly written in the manuscript. Based on my reading of the section "Does LLCPS apply to other silk types?", it seems like the authors mean to suggest that the polyalanine is neither a sticker or a spacer but should be considered separately as not part of the LLPS domain.

We fully agree, the division of the repetitive region into stickers and spacers was not clearly described by us. The reviewer is right, our intention is to separate the LLPS-forming stickers and spacers from the zipper-forming poly-A regions. Stickers and spacers are found in the flexible regions that connect the poly-alanine segments. They contain R/K and Y residues, as canonical stickers mediating LLPS, and G and S as spacers. Alanine is indeed not a sticker, and poly-A is not particularly flexible, but has a high propensity to form steric zippers. The repetitive region of MaSp1 could be summarized as "stickers, spacers and zippers". We have clarified this in the first paragraph of the section "Does LLCPS apply to other silk types?".

Figure 1 shows the general model for stickers and spacers, not for repetitive regions in particular. We have changed the sequence of the spacers to include serine and not only alanine, to avoid confusion

2) In line with my comment #1, the mapping between stickers/spacers (LLPS theory) and hard(crystallizing)/soft repeating domains (LC theory) is still not clear.

In our manuscript, we suggest the liquid crystal state connects the liquid droplet state of spidroins during storage and the crystalline state of the spidroins in the silk.

The exact nature of the mesogen (the liquid crystalline part) of silk proteins is a matter of debate, but for silkworm proteins, it has been suggested to be micelle-like structures (Reference 52, Walker, Holland, and Sutherland, Proc Royal Soc B, 2015). Spidroins assemble into similar micelle-like structures that are likely kept from aggregation through LLPS mediated by stickers and spacers in the repeat domains. These structures become more and more ordered during spinning, for example by crosslinking via the N-terminal domains, giving rise to the mesogen that accounts for the LC properties of the dope. Inside the isotropic mesogens, the crystal-forming sequences (polyA) assemble into liquid crystal microdomains. We have clarified our proposed model in the text, and put it into the context of the model by Walker et al., see the last two paragraphs of the section "Evidence for LLCPS spinning of

spider silk”, and the new figure panel 3d, which puts the silk phase transitions into the context of LC theory.

3) The authors refer to the polyalanines as ~15 residues long, with alpha-helical structure in the pre-spinning state. However, many polyalanine segments in spidroins are shorter than 15 residues. For example, ADF-4 has polyalanines that are approximately 6-12 residues long. At these lengths, the segments have less helical propensity and more random coil propensity (<https://pubs-acrs-org.libproxy.rpi.edu/doi/10.1021/bi201155g> and others). This discussion should be added to the manuscript.

Thank you for bringing this aspect to our attention! We agree, shorter poly-A segments have a lower helical propensity. They do still form fibrillar aggregates (see the study by Bernacki and Murphy referenced above). We have modified the text to clarify that alanine repeats in spidroins vary in length and secondary structure, and added the suggested reference, as well as a reference to the polyalanine length in MaSps.

4) Lines 198-199 describe phenylalanine as a lower affinity sticker compared to tyrosine, which increases LLPS. It's not clearly explained why phenylalanine less sticky than tyrosine, and why lower stickiness increases LLPS. Similarly, it's not clear to me why Leu is a higher affinity sticker.

The section in question can indeed be explained better. Our statements refer to the work by Tanja Mittag and co-workers who have mapped the contributions of individual amino acid stickers to LLPS and found that tyrosine and arginine are the strongest promoters of phase separation. Based on their ranking, we mutated Y and R sticker residues in NT2RepCT and found that weaker stickers (F instead of Y, L instead of R) increase the size and fluidity of spidroin droplets (Leppert et al, Nano Lett 2023). We have clarified this in the text and the legend to Figure 2.

5) On line 232, the authors speculate that "LLPS helps to keep the poly-alanine regions from spontaneous alignment". However, it's not clear to me why polyalanine regions would spontaneously align. This may be a wording choice. Polyalanine segments may spontaneously form beta-strands that are locally aligned on the angstrom scale, but without shear or elongational flow, they should not universally align along any particular axis.

Yes, “align” is not a good wording choice. We refer to the tendency of poly-alanines to spontaneously self-associate and subsequently aggregate. We have changed the wording to “We speculate that in the dynamic environment of a liquid droplet, self-association tendencies may be mitigated by LLPS of the linking regions between the polyalanines.”

6) On line 239, the authors state "Hence, the pre-spinning state might be described as LLCPS...". This is a bit unclear to me, as no crystallization occurs (according to the authors on lines 234-235) until the dope enters the spinning duct.

We suggest that the travel from sac to distal end of the duct is accompanied by a shift from LLPS, to LC, to crystalline state, so the reviewer is correct that “pre-spinning” is unclear. We have removed the term.

7) Lines 181 - 183 state that "Further studies are therefore needed to confirm that both liquid-crystalline and liquid-liquid states co-exist under spinning conditions". These studies seem like they should be straightforward, so a discussion of why such studies have not been performed (challenges and barriers?) would enhance the manuscript. Additionally, more discussion on

what is known about the liquid crystalline states, and how that may conflict or coincide with LLPS states would be helpful. For example, how might a nematic liquid crystal phase "fit" into a spidroin-rich LLPS domain? Presumably, LC domains would be much smaller and therefore reside within the LLPS domains?

The studies confirming liquid crystalline properties in silk dope predate much of the high-resolution structure information we have today, as well as the availability of recombinant mini-spidroins that recapitulate features of native spidroins. To us, the next key question is whether LLPS and liquid crystalline properties are related, which can be addressed by structure analysis of mini-spidroins under different assembly conditions.

Regarding the role of a nematic liquid crystal phase, we speculate that the beta sheet-forming sequences assemble inside the mesogen micelles that form through LLPS. A recent study by Linder and co-workers shows that spidroins inside droplets get trapped in dynamic arrested states, which would agree with the formation of micro-crystals by the poly-alanine repeats (Fedorov et al, *Adv Funct Mater*, 2410421, 2024). Such micro-crystals could align while still embedded in the liquid phase when traveling through the duct (see e.g. De Luca & Rey, <https://doi.org/10.1063/1.2186640>). Therefore, studies on how the assembly of the alanine repeats is affected by mutating residues that drive LLPS, as well advice versa, would be a key puzzle piece. We have added these points to the discussion of possible future directions at the end of the conclusion section.

8) on line 142, the authors state that the polyalanines "account for fiber strength". This seems to be an oversimplification, as the beta-sheet nanocrystals also contribute to stiffness and toughness. (<https://www.nature.com/articles/nmat2704> and <https://doi.org/10.1002/adfm.201600813> may be good references)

We agree, we have changed the term to "contribute to fiber strength" and included the suggested references.

Reviewer #3 (Remarks to the Author):

This review is well-organized and good at illustrating. I recommend the publication of this review after minor revisions:

We thank the reviewer for the positive assessment!

1. In Figure 1, I suggest including both interchain interactions (such as anti-parallel β -sheets) and intrachain interactions. This will provide a more comprehensive understanding of the structural dynamics during silk assembly.

We have included this suggestion in the modified Figure 1. Note also our response to Point 1 by reviewer 1.

2. It is beneficial to offer a more detailed illustration of the dimerization of the N-terminus through electrostatic interactions.

We have included an illustration in Figure 2.

3. What are the specific ion species mediating spider silk glands' phase transition? This could help clarify the molecular mechanisms involved in the transition from LLPS to a solid state.

In the sac, the concentrations of Na⁺, K⁺ and Cl⁻ is 192, 6 and 164 mM, respectively (Andersson et al. *PLoS Biol.* 2014). The exact concentrations of the gradients are yet

unknown, but groundbreaking work by Vollrath and co-workers has shown that potassium, phosphate, and sulfate are increased, while sodium and chloride are decreased (see Knight and Vollrath, *Naturwissenschaften* 88, 179-181, 2001). We have added this information to the description of the spinning process.

4. On page 6, the author summarized factors influencing the state of spidroins, stating that "The state is determined by factors like concentration, ionic strength, and temperature, depending on..." I recommend adding pH and shearing force to this list, as these are also critical parameters that could influence phase transitions during silk formation.

We thank the reviewer for the suggestion and have included shear and pH, which are indeed the most critical factors.

Reviewer #4 (Remarks to the Author):

The structured, well-written paper is of high interest to other researchers in the silk community, but also for the wider field of (fibrous) biopolymers.

Thank you for the encouragement!

However, there are several aspects in the manuscript that need to be addressed or clarified:

1) The main concept of the authors is the combined view of LLPS and liquid crystal state of the silk spidroin, leading to the proposed LLCPS behaviour. This is mainly introduced in the paragraphs 1.163ff & 225ff, and the way the authors formulate their arguments could lead to a conceptual misunderstanding:

A) crystal and crystallinity aspect: the crystalline β -sheet formation of Poly-A stretches is not directly related to a liquid crystal state of a (bio)polymeric material (see also abstract), since the latter is defined as a bulk material state with specific properties, due to ordered molecules, or mesogens, that display liquid-like behaviour. Upon reading the manuscript, the reviewer struggled with the proposed view that β -sheet formation of the Poly-A stretches as crystallization leads to the liquid crystal state of the dope.

B) Conceptual and sequential aspect: As cited, liquid crystal state has been proposed for silkworm silk by Walker et al. 2015; they argue it is based on micelle-like lyotropic mesophases, responsible for the orientation and properties of the dope. Thus, the micellar-like structures themselves seem to lead to liquid crystal state/behaviour, and "Solidification occurs concurrently with the structural transition to the final extended- β -sheet structure". Thus, shear stress, pH-induced conformational changes, ion conc. & composition changes, and increased intermolecular interactions in concert would lead to LLCPS and the formation of solid fibers as final result.

However, the reviewer is not questioning the proposed general concept of the LLPS and LLCPS behaviour of the silk spinning dope. The authors are asked to clarify these aspects/rewrite text passages for better understanding.

We agree that these distinctions need to be clarified and apologize for the confusion. We suggest that silk is in an LLPS state which acquires LC characteristics. When spinning is initiated, the micelle-like structures inside the liquid droplets take on the role of the mesogens as the dope is sheared and concentrated. On an interesting side note, the well-established pH-dependent dimerization of the NT domain may help to organize the micelle-like structures, and thus promote the shift from LLPS to LC characteristics. The polyalanine assemblies that form inside the mesogens represent microdomains (or tactoids) that give rise to the nematic

phase, and eventually become the crystalline component that shifts the silk dope from LC to crystalline, spun silk. We have clarified this in the last two paragraphs of the section “Evidence for LLCPS spinning of spider silk” and in Figure 3.

2) Conceptual confusion or wording issue using the term aggregation in the manuscript: the reviewer disagrees with the view of Poly-A aggregation to β -sheet secondary structures. Instead, the term assembly should be used throughout, since protein aggregation is def. as spontaneous, uncontrolled structure formation. This applies even more so to aggregation of the CT-domain I.236 and Fig. 3. In the cited references e.g. Hagn et al. 2010, the pH-sensitive CT acts as a conformational switch due to exposure of hydrophobic domains upon acidification, allowing the two rep. core regions to get closer together and alignment of β -sheet forming repetitive sequence elements. There is no evidence of CT aggregation, but the dimer remains stable. It is highly unlikely that the concerted fiber spinning assembly incorporates uncontrolled aggregation mechanisms.

We agree that “aggregation” invokes the image of an uncontrolled process. We have changed the term to “self-assembly”, with the exception of the aggregation hot-spots predicted by FuzDrop. We would like to point out that several studies published after Hagn et al. have shown that low pH destabilizes the isolated CT dimer and induces the conversion to β -sheet-rich assemblies (see e.g. Andersson et al, PLoS Biology 2014; Li et al, Biomacromolecules 2022; Rat et al, Protein Sci 2023; De Oliveira, Nature Commun 2024, and others).

3) The role of ion conc. and composition is generally reflected in the manuscript, but it seems that the authors focus more on pH shift and shear stress as spinning factors, and reduce the role of ions in their text mostly to ion concentration changes. The role of ion exchange, i.e. composition of chaotropic and kosmotropic ions, has been established in the literature, and should be included in more detail in the manuscript to add clarity.

Thank you for the suggestion. It feels like the silk community is more focussed on shear and pH since these are the better-understood factors, but the switch from chaotropic to kosmotropic ions is likely evenly relevant. We have included the additional information in the discussion of the assembly process.

4) Another conceptual confusion or wording issue using the term micelle in the manuscript: micelles usually describe solvent-driven spherical structures with double-lipid layer. The silk community should use micelle-like, or micellar, structures of the spidroins instead for clarity.

We fully agree. The term “micelle” was used by us since it is rooted in literature going back close to 20 years. We have changed the wording to “micelle-like” throughout.

5) In the paragraph on other silk types, it would be helpful if the authors may include both a more detailed discussion of the abundance of so-called “aggregation hot-spots” (presumably β -sheet forming motifs) in relation to the discussed LLCPS behaviour and fiber formation; in addition, it would be interesting to further discuss the reported results in relation to the various biological functions of the diff. silk proteins. It would be particularly interesting to discuss the differences in the two pyriform spidroins, as well as the consequences of the LLPS or LLCPS behaviour for this attachment cement.

We have included a more detailed description of “aggregation hot-spots” as defined by FuzDrop.

Regarding other silk types, the reviewer raises a very interesting issue, namely the assembly of the non-dragline silks. Open questions relate to differences in the amount that needs to be produced, the requirements regarding spinning speed, and the final structure of the spun fibers between dragline, attachment, and coating silk fibers. We mention some of this in the section on other silk types, however, there is considerably less data, and we feel that the discussions the reviewer suggests will be too speculative.